# FEM Analysis of Individualized Polymeric 3D Printed Guide for Orthodontic Mini-Implant Insertion as Temporary Crown Support in the Anterior Maxillary Area

**DOI:** 10.3390/polym15040879

**Published:** 2023-02-10

**Authors:** Riham Nagib, Andrei Zoltan Farkas, Camelia Szuhanek

**Affiliations:** 1Orthodontics Research Center ‘ORTHO CENTER’, Department of Orthodontics, ”Victor Babes” University of Medicine and Pharmacy Timisoara, Eftimie Murgu Sq. 2, 300041 Timisoara, Romania; 2Department of Mechatronics, Polytechnic University of Timisoara, 1 Mihai Viteazu Ave., 300222 Timisoara, Romania

**Keywords:** orthodontics, polymeric resin, mini-implant guides, 3D printing, FEM analysis

## Abstract

Either due to trauma, extraction or congenital factors, the absence of teeth has aesthetic, functional, financial and psychological consequences. The aim of the current study is to assess an individualized polymeric 3D printed digitally planned surgical guide designed to achieve precision and predictability in non-standard mini-implant orthodontic cases. Twenty-seven patient records with missing anterior teeth were selected from the database of a private clinic in Timisoara, Romania. Based on the analysis of the cases included in the research, a surgical guide for the insertion of mini-implants as provisional crown support was designed. An FEM simulation was performed using the Abaqus numerical analysis software. Finite element simulation revealed the maximum displacements and stresses that occur in the surgical guide. Mini-implant supported provisional crowns can be a simple and low-cost method to increase patient self-esteem and compliance with the orthodontic treatment. Computer aided mechanical simulation is a useful tool in analyzing different polymeric surgical guide designs before being used in clinical situations in order to avoid failure.

## 1. Introduction

The absence of maxillary incisors is a frequent motivation for orthodontic treatment. Tooth agenesis is one of the most common dental anomalies of the last decades with reports related to prevalence increasing as dental imaging technology advances and patients are becoming more aware of their dental status [1]. In the permanent dentition, third molars are the most common missing teeth, while the second most common are either the lower second premolars or upper lateral incisors [2,3]. Lower lateral incisor agenesis is less frequent when compared to mandibular central incisor agenesis incidence [4]. Differences in prevalence have been correlated with racial group, geographical location, orthodontic and non-orthodontic patients, genders and genetic syndromes [5,6,7,8,9,10]. Another reason for the absence of frontal teeth is trauma which has been observed in both primary and permanent dentition and irreversible damage leading to ankylosis, resorption or periodontal conditions could eventually require the affected teeth to be extracted [11]. Avulsion and decay processes are also factors in tooth loss in the maxillary incisor area.

Although anchorage control is the main use of mini-implants in orthodontics, studies have reported successful use of these devices as support for provisional crowns as an alternative to bonded bridges and dentures, in both growing and adult patients [12,13]. Being one of the most prevalent missing anterior teeth, the majority of studies describe this method for the replacement of the upper lateral incisor [14,15,16,17,18].

Static guiding systems transferring predetermined implant sites using surgical templates in the patient’s mouth have been increasing the success rate of mini-implant insertion in orthodontics [19]. In free-hand surgery, 2D or 3D radiographs are usually used to assess the alveolar bone available to place the implant and examine the surrounding anatomical structures. The intraoral examination is also a part of the planning stage with periodontal probes or calipers being used to provide the surgeon with a view of the height and thickness of the ridge [20]. The adjacent teeth can be used as a guide for determining the correct insertion site and position of the mini-implant. Poor surgical technique, mistakes in treatment planning, insufficient space and incorrect angle can lead to damage of the periodontal ligament and surrounding structures. This can lead to changes in blood supply of surrounding teeth, apical periodontitis and root resorption. Incorrect positioning of the implants can lead to implant failure due to bone resorption or infection and also irreparable damage to adjacent teeth [21].

Scientific clinical evidence concerning the precision of 3D guided insertion of mini-implants for temporary orthodontic anchorage has increased in the last decades due to an abundance of papers on different surgical guide designs and research on their benefits to mini-implant insertion procedures [22,23,24,25,26,27,28]. Analog surgical guides have also been described and have delivered satisfactory results [29,30], but due to the technological advancements of recent years, computer aided planning and design is now the preferred method in this field. The digital protocol for the computer-assisted static orthodontic mini-implant insertion procedure involves cone beam computed tomography of the patient, and either an intraoral scan or a laboratory scan of the study model. Superimposition of the data is required so that implant planning can be done virtually using computer software. The plan is then transferred to the patient using surgical guides that direct the drill or the hand instrument [31]. The concept of computer-guided implant surgeries has numerous benefits in the clinical scenario, minimizing risks and complications and surgical guide designs have evolved from bone borne to mucosa or tooth borne, from conventional planning to digitally planned, from milled to 3D printed, and from incorporating metal inserts to fully polymeric guides [32]. 

Considering the characteristics of the insertion site and the needed precision, surgical guides can be classified in four types: tooth supported, tissue supported, tooth and tissue supported, and tissue supported with an accessory for edentulous cases [33]. Based on the amount of limitation that the surgical guide offers, three different concepts have been described in the literature: nonlimiting, partially limiting and completely limiting design [34]. In less limiting designs, the operator has the freedom to decide on parameters related to the final angulation and depth of the implant, while restrictive designs decrease operator decision-making during the surgical procedure.

Given the wide range of template design characteristics, studies have shown FEM simulation to be a reliable method that can be used to observe and find solutions to a wide range of problems including stress analysis and has been successfully implemented in the assessment of polymeric surgical guides [35]. Still, there are limitations to the use of the guided implant insertion technique concerning the range of the mouth opening and insertion site [36]. Many studies support the fact that pre-operative digital planning and the use of static guiding techniques allow for accurate and controlled placement of orthodontic mini-implants while minimizing risks associated with the clinical procedure [37,38]. 

Polymer additive manufacturing technologies currently available are a reliable option in dentistry [39]. The use of 3D printing in orthodontics focuses mainly on study model fabrication, custom made appliances, occlusal guards, clear aligner printing, drilling templates and surgical guides for mini-implant insertion [40,41,42]. Other uses include tooth auto-transplantation, guided osteotomy, piezoelectric corticotomy and root resection [43]. Advances in resin material and 3D printers provide the means of obtaining accurate biocompatible surgical guides [44,45]. Even so, material behavior studies are still needed to provide insight into clinical situations [46,47]. 

The aim of our study is to assess the need for and provide a means of simplifying the insertion of temporary anchorage devices (TADs) as support for aesthetic reconstructions in the anterior dental area, by creating a tooth borne open frame individualized 3D printed polymeric guide and simulating its behavior in a clinical setting. 

## 2. Materials and Methods

### 2.1. Statistical Study

To assess the need of polymeric individualized surgical guides in orthodontic practice, the database of a private clinic in Timisoara, Romania, was used to collect orthodontic records of patients with missing teeth in the maxillary frontal area out of a total of 235 patients from 2018 to 2021. All records contained a signed consent form and patient information sheet. The inclusion criteria were high quality of orthodontic records (panoramic radiograph and/or maxillary CBCT) and absence of upper central or lateral incisors. The records had uniform characteristics and were examined by a single operator. Syndromic patients were excluded from the present study. 

The study sample consisted of a total of 27 cases, 19 females and 8 males of ages between 10 and 37 years. Data obtained from patients’ records were organized according to gender, age, cause of tooth loss, type, location and number of missing teeth. Bone resorption and amount of lost space on the arch were also assessed. Data were included in a Microsoft Excel sheet. Descriptive statistical methods were performed. 

### 2.2. Computer Aided TAD Insertion Guide Design

Taking into consideration the data obtained through the first part of the current research, a polymeric surgical guide suitable for the placement of mini-implants in the frontal maxillary area was conceived. The features of the design permitted direct visualization of the insertion site and static guidance of the manual instrument, thus providing an easier surgical intervention (Figure 1). The base of the guide was custom made to fit the chosen case, but could be easily modified to fit any other situation. 

A random cone beam computer tomography (CBCT) from the study sample taken with CRANEX 3D (Soredex, Tuusula, Finland) equipment was used for the design stage of the insertion guide. Implant planning consisted in deciding the angle of insertion for an 8 mm long and 1,6 mm diameter orthodontic mini-implant (Figure 2). Computer aided design (CAD) software Exocad (exocad GMBH, Gerlingen, Germany) provided the tools to create the surgical guide. 

The final design was exported for printing with biocompatible polymer resin “Surgical Guide” (Vertex-Dental, Soesterberg, Netherlands) to the Form2 (Formlabs Inc., Somerville, MA, USA) 3D printer. The printed result was rinsed twice in an ultrasonic bath with alcohol solution (96%) in order to remove any excess material. The parts were dried and placed in a UV light curing box for final polymerization. All support structures were removed and the guide was polished (Figure 3).

### 2.3. Finite Element Analysis of Individualized TAD Insertion Guide

FEM is a numerical method that can be used to find the solutions to a wide range of engineering problems such as stress analysis, heat transfer analysis, electromagnetic field analysis or fluid flow analysis. 

The method is based on the idea of constructing complicated objects from simpler objects, or the division of complicated objects into simpler objects, called finite elements (FE), for which known calculation schemes can be applied. The operation of dividing the real model into parts of small dimensions is called meshing [48]. 

The orthodontic guide obtained in the previous stages of our study was imported into the CATIA R5V19 (Dassault Systems, Vélizy-Villacoublay, France) software where it was processed in order to transform it into a solid 3D file, a file necessary for importing into the Abaqus FEM analysis software. The CATIA Part solid file was imported into the Abaqus FEA (Dassault Systems, Vélizy-Villacoublay, France) analysis software.

A material was created with the material editor integrated in the analysis software. Physical and mechanical properties of the biocompatible polymer resin “Surgical Guide” (Vertex-Dental, Soesterberg, Netherlands) were used, namely a uniformly distributed density of *ρ =* 1.15 × 10^−9^ tonne/mm^3^, Young’s modulus of *E =* 2900 MPa and a Poisson’s ratio of *ν =* 0.3, according to the technical data provided by the resin manufacturer [49].

After application of the material properties to the imported surgical guide file, the model was meshed using an approximate global seed size of 1 (Figure 4a). On the surface of the printed model that came into contact with the patient’s occlusal plane, an “encaster” boundary constraint was created to simulate the contact with the teeth (Figure 4b). In the bore of the actual guide (insertion site of the orthodontic TAD screwdriver), a “Multi-point” type beam constraint was created to simulate the body of the screwdriver used in fixing the mini-implant (Figure 4c), on which we applied a concentrated force with a value of 50 N on the *y*-axis (Figure 4d), a value equivalent to the maximum force applied by the hand of a human operator [50] in cases of an improper maneuver while using the surgical guide.

After all the material, load and constraint parameters where set, a job was created in Abaqus’s Job Manager and a full analysis was submitted.

## 3. Results

### 3.1. Statistical Study

The final record collection comprised 27 patient records with 19 (70.37%) females and 8 (29.62%) males and a mean age of 23.50 ± 13.50 years. All patient records contained a digital panoramic radiograph. Seven cases also presented a maxillary CBCT. A prevalence of 11.48% was reported for missing maxillary incisors in the total group.

Patient motivation for orthodontic treatment was reported to be ‘esthetic issues’ by 92.50% of patient records. Tooth loss causes mentioned in the analyzed files were extraction due to decay or unsuccessful dental procedures in 18.51% of cases, traumatic avulsion 3.70%, complications due to previous trauma 25.92% and tooth agenesis in 51.85% of considered cases. Total number of missing teeth was 36. No significant difference was noticed between the left and right side of the arch. The majority of cases displayed unilateral upper lateral incisor agenesis 31.55%, followed by bilateral upper lateral incisor in 20.30% of studied cases. One case had lost all four maxillary incisors. Absence of both central incisors was noticed in 13.92% of cases. Combinations of maxillary central and lateral incisor loss were observed in the remaining cases. No case of central incisor agenesia was reported. Space loss was noted in all cases to a certain degree, but mainly in the agenesis group, while bone resorption was more evident in trauma or extraction cases. Results of this section of the study are detailed in Table 1.

### 3.2. Finite Element Analysis of Individualized TAD Insertion Guide

Finite element simulation revealed that maximum displacements occur in the area of the guide cylinder, with a maximum displacement value of 0.297 mm (Figure 5a). The maximum Equivalent Stress was recorded at the junction between the guide support and the guide base. A maximum value of 95.806 MPa was observed (Figure 5b).

To plot the stress distribution on the deformed structure, we created a contour on the deformed shape (in the Visualization module) to obtain a more accurate data reading regarding the stresses and deformations in that area. To obtain the stress variation along the critical zone we defined a path of nodes with the concentrator tip (the point of maximum stress) somewhere in the middle of the contour (Figure 6).

After selecting the path of interest, a graph illustrating the stress variation in each of the selected nodes was plotted (Figure 7).

The Equivalent Stress, as seen in the plotted graph, does not have any significant values along the path of nodes other than in the point of maximum stress that recorded a value of 95.806 MPa.

## 4. Discussion

The epidemiological investigation of missing teeth in the anterior area determined the prevalence rate of these cases and thus the need for provisional reconstruction on mini-implants during or after orthodontic treatment. As other studies reported, the majority of orthodontic patients are female [9]. In our study, 70.37% of our sample was female patients. Taking into consideration that the records were provided by an orthodontic clinic, results might not characterize the general population. A prevalence of 11.48% was reported for missing maxillary incisors in the total group, similar to results found in the literature [5,6,7,8]. Unilateral upper lateral incisor agenesis was recorded in 31.55% of cases but no significant difference was noticed between the left and right side of the maxillary arch. Successful treatment of this anomaly often requires a multidisciplinary approach to establish both optimal esthetics and functional occlusion either by considering space closure and canine substitution of the lateral incisor, or by space opening [2]. Regaining the space for replacement of the missing tooth has an undesired effect on smile aesthetics and can negatively impact patient cooperation [1].

Data showed mention of undesirable smile and aesthetics as main complaint in 92.50% of orthodontic records. This provided the basis for the conception of an easily customizable way to provide support for temporary reconstruction of the missing teeth during the orthodontic treatment and post-treatment until the final restoration is in place. This method is recommended to temporarily satisfy the esthetic needs of the patient and can be used as a space maintainer. Mini-implant prosthetic rehabilitation is a viable option for provisional rehabilitation of growing patients because it preserves the bone structure while restoring function and esthetics until the end of growth [13].

Many orthodontists avoid using mini-implants due to fear of failure and limitations of surgical skills [12]. Providing a more accurate alternative to free hand insertion could encourage clinicians to take this method into consideration. The temporary replacement of missing maxillary lateral incisors has been followed up with no notable side effects reported [14,15,16]. Most of the orthodontic mini-implants do not osseointegrate and can be easily removed at any point and replaced with a dental implant [18].

Although many studies report using static guidance in the insertion of orthodontic mini-implants, the surgical guide characteristics are mostly similar [22,25,26,27,28,29,30]. The concept described in the present study focuses on the anterior maxillary area of orthodontic patients with missing frontal teeth. The novelty of the design lies in the fact that it was conceived after a comprehensive study of the particularities of the aforementioned cases. The open frame concept used in our design has proven to have clinical advantages in guided dental implant surgery [32,51]. The guide presented in this study has a modified sleeve that guides the mini-implant manual insertion instrument that minimizes the space needed for the correct positioning of the guide, both vertically and horizontally. Conventionally, the metallic or polymeric sleeve is positioned immediately above the insertion site, over the soft tissue [27,28]. In both dental implantology and orthodontic mini-implant placement this fact requires the sleeve to be adjusted to the different surgical kits and mini-implant insertion tools available on the market [40,52].

Firstly, raising the sleeve above the occlusal plane, as in the design presented in this study, allows a better positioning of the surgical guide given that when space between adjacent teeth is limited surgical guide insertion can be difficult [52,53,54]. Driving the hand instrument avoids errors that can appear when moving the guide sleeve away from the insertion site [51]. Lack of space between neighboring teeth and bone resorption in the anterior area are consequences of missing maxillary teeth, being present to some degree in the whole study sample. These characteristics influence the construction of the surgical guide. According to several studies, the dimensions of the guiding sleeve significantly influence the precision of the procedure in dental implant surgery as well [55]. A height less than 5 mm is not actually able to guide the preparation of the implant site as planned [56,57]. To increase accuracy, the guiding cylinder on the presented surgical guide was designed to have a height of 10 mm.

Secondly, the open field concept in which the mini-implant does not come into contact with the structures of the guide solved the problem of contamination of the implant surface which can be a cause of implant failure [41,42].

Thirdly, the design offers the clinician perfect visual control during the insertion procedure. Orthodontic mini-implants do not require a soft tissue flap or drilling in most cases [40,41,42]. Specific aesthetic reconstruction reasons should take into consideration the relationship between the mini-implant and the gingiva. In the conventional surgical guide designs, the operator cannot see the insertion site and thus cannot properly manage the surrounding soft tissues [51,52,53].

The concept can be applied for the insertion of temporary anchorage devices as support for aesthetic reconstructions in the anterior dental area and the proposed polymeric static guidance device, even though not universal, but can be easily customized to fit any specific circumstance. The cylinder that guides the manual insertion instrument does not need any modification if the instrument remains the same, regardless of whether the diameter or length of the chosen mini-implant changes. The base resting on the teeth and the angulation of the guiding cylinder need to be modified according to the characteristics of each individual case. Although free-hand when compared to the guided technique of mini-implant placement eliminates the time required to digitally plan the guide and reduces the cost of printing, the advantages of using surgical guides overcome these limitations [20,21]. Free-hand insertion requires a longer clinical procedure given that the clinician is planning the position of the implant on the spot. Thus, the risk of human error is higher with less predictable results which can cause complications for the patient (infection, implant failure, incorrect positioning, swelling, pain) [42]. Studies have shown that guided insertion addresses all the above risks, decreasing surgery time and providing more accurate results [20,24,27,28]. The application of this protocol has great advantages in nonstandard cases like the one described in the present study (space loss, singular missing teeth, aesthetic area, alveolar crest resorption). This method is also recommended for beginner clinicians due to the high success rate and reduced failure, which could cause higher costs than the cost required for surgical guide fabrication [20]. No studies have addressed the comfort level of patients during these procedures, but provided that the guide simply rests on the neighboring teeth, this should not be considered a limitation. Surgical guide printable resin materials are suitable for use in dentistry and for sterilization at 137 °C which studies report to improve accuracy and overall polymerization. [28,58]

Data analysis of the FEM simulation revealed that the design concept chosen for the surgical guide can resist unforeseen forces that may appear during its use, well within the parameters imposed by the resin manufacturer, which mentions a value of 103 MPa for the resin’s flexural strength [49]. The construction of the analyzed surgical guide was based on a specific case. Therefore, predictable and accurate clinical results can be expected when using this design for the surgical guide. Considering that the maximum value obtained in the simulation was 95.806 MPa in an FE of the mesh and that the rest of the analyzed area registers equivalent stresses under the value of 50 MPa, the researchers consider that this surgical guide design is validated for clinical use by means of FEM analysis. Maximum displacements occurred in the area of the guide cylinder, with a maximum displacement value of 0.297 mm. The literature describes clinically acceptable error limits within a range varying from 0.13 to 0.57 mm [37]. Despite possible clinical irrelevance, potential sources of errors in the manufacturing process of the polymeric guide should be carefully addressed. Successful simulation of a clinical procedure using the surgical guide design concept presented in this study is just a preliminary stage and this should be considered a limitation of the current research.

## 5. Conclusions

The statistical data analysis concluded that orthodontic patients with missing teeth in the frontal maxillary area have a high prevalence; they require temporary aesthetic reconstruction of the area and display unique characteristics of the alveolar bone. These conclusions led to an individualized polymeric 3D printed surgical guide design concept meant to aid in the application of orthodontic TADs as provisional crown support. Taking into consideration the limits of the current research, the proposed concept can easily overcome the difficulties of surgical guide designing and mini-implant insertion in nonstandard situations like the cases discussed in the epidemiological study. The conclusions of the computer aided FEM analysis, used as a tool in analyzing different 3D printable polymeric surgical guide materials, are that the guide design presented clinically irrelevant displacements and stress levels within safe ranges for the extreme clinical situation simulated. Thus, this guide design can be used in order to avoid failed or inaccurate procedures. More study designs (different operators, implant position assessment, clinical studies) are needed to gather the data needed to definitively validate the presented solution for the simplification of mini-implant insertion when used as support for provisional crowns in the anterior maxillary area, increasing patient self-esteem and compliance with the orthodontic treatment.

## Figures and Tables

**Figure 1 polymers-15-00879-f001:**
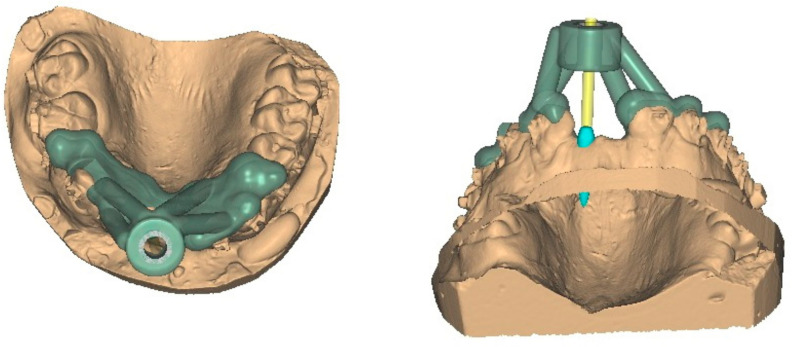
Individualized 3D printed surgical insertion guide for mini-implants in the maxillary frontal area. The blue component is the simulated TAD. The yellow component simulates the insertion tool (orthodontic mini-implant screwdriver).

**Figure 2 polymers-15-00879-f002:**
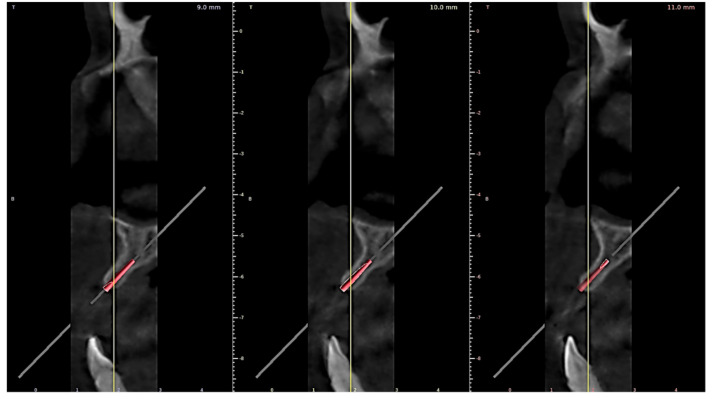
Insertion angle planning: 3D virtual model of the orthodontic mini-implant viewed in saggital plane sections of the CBCT investigation.

**Figure 3 polymers-15-00879-f003:**
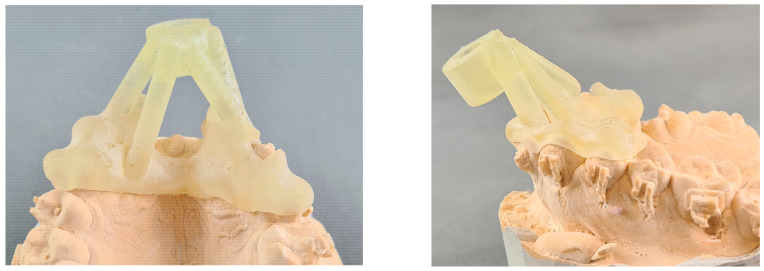
Three-dimensional printed surgical guide for orthodontic placement of orthodontic TAD to support provisional crowns in the esthetic area.

**Figure 4 polymers-15-00879-f004:**
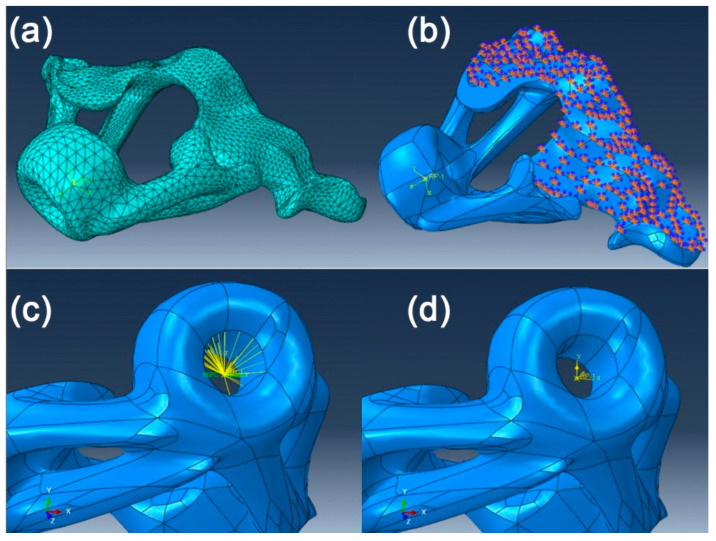
(**a**) meshed surgical guide model; (**b**) applied boundary constraints; (**c**) MPC constraint boundary condition; (**d**) applied load on the surgical guide MPC constraint.

**Figure 5 polymers-15-00879-f005:**
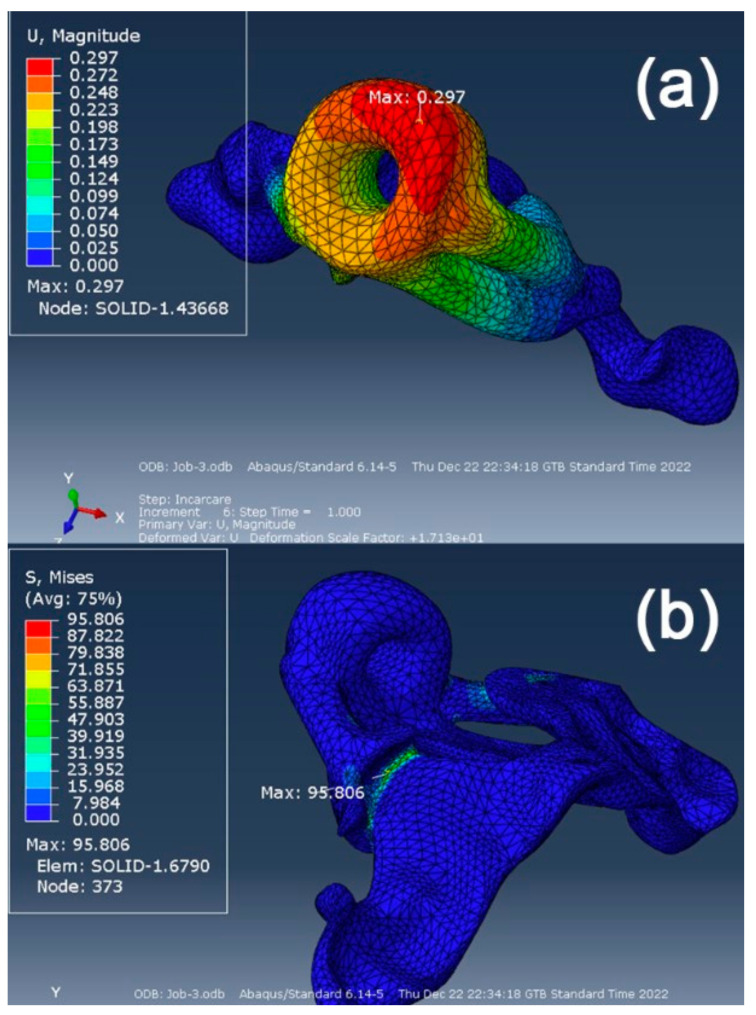
(**a**) maximum displacement and (**b**) equivalent stress results.

**Figure 6 polymers-15-00879-f006:**
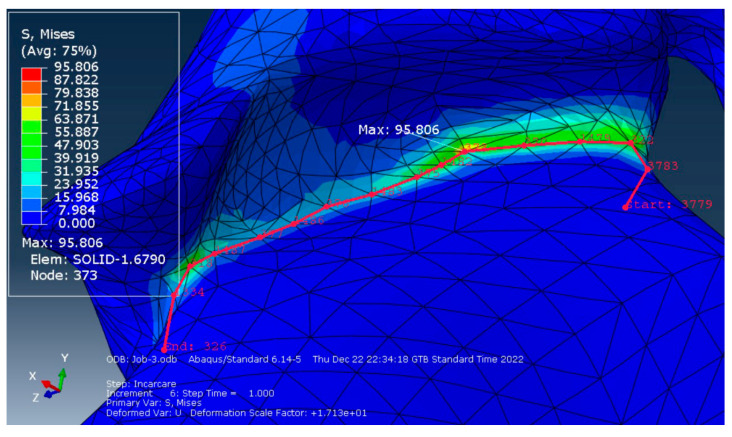
The path of nodes surrounding the point of maximum stress.

**Figure 7 polymers-15-00879-f007:**
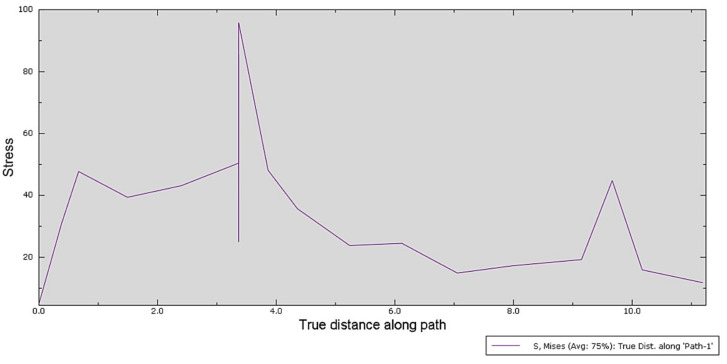
Stress variation along the critical zone, Path nodes (*x*-axis) and the Von Mises stress field output (*y*-axis).

**Table 1 polymers-15-00879-t001:** Prevalence of maxillary incisor loss or agenesis.

	Minimum Prevalence	Maximum Prevalence
Number of missing teeth		
	3.70%	31.55%
	4 missing incisors	1 missing incisor
Cause of missing teeth	3.70%	51.85%
	avulsion	agenesis
Type of agenesis	20.30%	31.55%
	Unilateral upper lateral incisor	Bilateral upper lateral incisor
	0%	0%
	Unilateral upper cental incisor	Bilateral upper central incisor
Gender	27.35%	73.65%
	male	female

## Data Availability

The data presented in this study are available upon request from the corresponding author.

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
