# Peer review of "FEM Analysis of Individualized Polymeric 3D Printed Guide for Orthodontic Mini-Implant Insertion as Temporary Crown Support in the Anterior Maxillary Area"

_polymers, 2023, doi:10.3390/polym15040879_

Round 1

Reviewer 1 Report

The manuscript "FEM analysis of individualized polymeric 3D printed guide for orthodontic mini-implant insertion as temporary crown support in the anterior maxillary area" has been reviewed. It deals with a FEM simulation using the Abaqus numerical analysis software. Simulation revealed the maximum displacements and stresses occurring in the surgical guide.

In my opinion the novelty aspects in relation to the "state of the art" must be better highligthed in the manuscript. In the present form the manuscript looks like a "technical report" rather than a research article.

Discussion and conclusions are too weak and should be enfatized, as coming from the main results. In particular conclusions are general sentences, not deriving from the results.

It should be deeply revised before resubmission.

Author Response

Statement 1:"In my opinion the novelty aspects in relation to the "state of the art" must be better highligthed in the manuscript. In the present form the manuscript looks like a "technical report" rather than a research article."

Answer: Upon drastic revision of the manuscript we found that the novelty aspects of the presented concept were indeed not properly highlighted, as you very well mentioned. We thank you for the recommendation and hope the present form of the paper is better suited for publication.

Statement 2:"Discussion and conclusions are too weak and should be enfatized, as coming from the main results. In particular conclusions are general sentences, not deriving from the results."

Answer: The discussion section was restructured and expanded. The conclusion section was rewritten altogether. We hope that in the present form these sections of the manuscript are stronger and emphasize our study's results.

Thank you very much for taking the time to review our work and for your on point recommendations.

Reviewer 2 Report

The evaluated manuscript entitled "FEM analysis of individualized polymeric 3D printed guide for orthodontic mini-implant insertion as temporary crown support in the anterior maxillary area" is interesting and well-crafted. The article provides new information on the orthodontic use of the mini-implant for clinical use. Illustrated illustrations and model photos are added to the manuscript. I have a few small comments about the text that need to be resolved:

1/ Table 1 etc. (throughout the text!) - incorrect decimal points used!

2/ The given article would perhaps deserve a slightly more extensive discussion with the addition of other recent sources.

Author Response

Statement 1: "Table 1 etc. (throughout the text!) - incorrect decimal points used!"

Answer: The issue was addressed. We thank you for pointing out this error.

Statement 2: "The given article would perhaps deserve a slightly more extensive discussion with the addition of other recent sources."

Answer: Both the state of the art and the discussion section were drastically revised and improved. Many recent sources have been cited and added as reference to our manuscript. We hope that in the present form you find our research better suited for publication.

Thank you for taking time to review our work and for your useful recommendations.

Reviewer 3 Report

The authors provide an interesting study examining the potential of a 3D-printed personlised guide frame for implant insertion in an orthodontic context. Briefly, the authors identified a number of cases from clinic where anterior teeth were lacking, and used the different cases to design a guide for implants in such contexts. The authors then utilised Finite Element Analysis to scrutinise the design and found there were very few points of concern in the design suggesting this may prove a valuable tool/approach in the field of orthodontics.

In reviewing the manuscript I made a couple of observations. The following should be considered by the authors when preparing a suitable revision.

1.       It is unclear whether the guide that was designed is universal or custom to a particular circumstance/case that was utilised as part of the study. The authors need to clarify this, and whether the FEM analysis applies to a singular guide design, or several.

2.       There are limitations to this approach in terms of time spent having the guide designed and ultimately printed, the ability to adequately sterile the material, how comfortable the guide is for the patient to interact with, etc. There are a number of translatable aspects not considered in recommending this approach that possibly warrant discussion.   

Author Response

Statement 1: "It is unclear whether the guide that was designed is universal or custom to a particular circumstance/case that was utilised as part of the study. The authors need to clarify this, and whether the FEM analysis applies to a singular guide design, or several."

Answer: We agree with your observation and in the new form of the discussion section we emphasized these aspects more thoroughly. To give you an on point answer, our concept is universal but for FEM simulation purposes we chose one of the cases in the epidemiological study and customized the surgical guide accordingly.

Statement 2: "There are limitations to this approach in terms of time spent having the guide designed and ultimately printed, the ability to adequately sterile the material, how comfortable the guide is for the patient to interact with, etc. There are a number of translatable aspects not considered in recommending this approach that possibly warrant discussion."

Answer: We thank you for this recommendation. We found it very useful in the revision process and all the mentioned issues and limitations are now widely debated in the discussions section.

We thank you for the time taken to review our work and the observations made to better revise the present study and make it more suitable for publication.

Round 2

Reviewer 1 Report

The manuscript has been significantly improved and can now be accepted in the present form.

Reviewer 3 Report

The authors have responded positively to my comments, and the manuscript is much improved.